# Collection of large benthic invertebrates in sediment traps in the Amundsen Sea, Antarctica

Minkyoung Kim[1], Eun Jin Yang[2], Hyung Jeek Kim[3], Dongseon Kim[3], Tae-Wan Kim[2], Hyoung Sul La[2], SangHoon Lee[2], and Jeomshik Hwang[1]

[1]School of Earth and Environmental Sciences/Research Institute of Oceanography, Seoul National University, Seoul, 08826, South Korea
[2]Korea Polar Research Institute, Incheon, 21990, South Korea
[3]Korea Institute of Ocean Science & Technology, Busan, 49111, South Korea

*Correspondence to*: Jeomshik Hwang (jeomshik@snu.ac.kr)

**Abstract.** To study sinking particle sources and dynamics, sediment traps were deployed at three sites in the Amundsen Sea for one year from February/March 2012, and at one site from February 2016 to February 2018. Unexpectedly, large benthic invertebrates were found in three sediment traps deployed 130−567 m above the sea floor. The organisms included long and slender worms, a sea urchin, and juvenile scallops of varying sizes. This is the first reported collection of these benthic invertebrates in sediment traps. The collection of these organisms predominantly during the austral winter, and their intact bodies, suggests they were trapped in anchor ice, incorporated into the overlying sea ice, and subsequently transported by ice rafting. The observations imply that anchor ice forms episodically in the Amundsen Sea and has biological impacts on benthic ecosystems. An alternative hypothesis that these organisms spend their juvenile period underneath the sea ice and subsequently sink to the seafloor is also suggested.

## 1 Introduction

The majority of the Amundsen Shelf in the Antarctic is perennially covered with sea ice, except for the two seasonal polynyas. The Amundsen Sea polynya in the west Amundsen Sea is the most productive polynya around Antarctica (Arrigo and van Dijken, 2003). Intensive flux of particulate organic carbon to the seafloor occurs in the austral summer while the sea interior is in starvation in the other seasons (Ducklow et al., 2015; Kim et al., 2015; Kim et al., 2019). Biogeochemical processes related to biological pump in the Amundsen Sea have been investigated by recent field campaigns (Arrigo and Alderkamp, 2012; Yager et al., 2012; Meredith et al., 2016; Lee et al., 2017).

Sediment traps were deployed in the Amundsen Sea to study sinking material flux and composition. Sampling occurred from February and March 2012 for one year at three locations along the paleoglacier-carved Dotson Trough, and from February 2016 to February 2018 in a trap deployed in front of the East Getz Ice Shelf, near Duncan Peninsula. Unexpectedly, at three locations macrobenthic organisms were found in the sampling cups, including long and slender worms, a sea urchin, and juvenile scallops of varying sizes. This is the first reported collection of these benthic invertebrates in sediment traps.

This paper reports on details of the benthic invertebrates collected, in addition to sinking particle flux data. The distinct environmental conditions and characteristics of the polar seas relative to temperate and tropical oceans probably explain the unusual occurrence of benthic invertebrates in sediment traps. For example, starvation in the winter due to a reduced supply of organic matter from the overlying water column may stimulate the relocation of benthos. The undersurface of the sea ice may provide a habitat for juvenile benthos before they settle to the seafloor. Anchor ice, which forms at the seafloor in supercooled water, can lift benthos to the overlying sea ice for further transport by ice rafting (Dayton et al., 1969). Studies in the Arctic have shown the incorporation of sediment particles and benthic organisms into sea ice, and their subsequent transport (e.g., Nürnberg et al., 1994). Potential transport mechanisms for the collected benthos are considered, and the implications of the surprising observations are discussed.

## 2 Methods

Time-series sediment traps (McLane PARFLUX Mark 78G; conical type, 80 cm aperture diameter, height/diameter ratio=2.5) were deployed on bottom-tethered hydrographic moorings during two cruises to the Amundsen Sea (Figure 1). The traps were positioned at four locations with different sea surface conditions: the sea ice region (Station K1); the central Amundsen Sea Polynya (Station K2); near the Dotson Ice Shelf (Station K3); and near the East Getz Ice Shelf (Station K4) (Figure 1). Station K1 (72.40 °S, 117.72 °W; water depth 530 m) was located in the sea ice region, near the northern entrance of a glacier-carved trough (Dotson Trough). A sediment trap was deployed here 130 m above the sea floor at a depth of 400 m, and collected samples from 7 March 2012 to 16 March 2013. Station K2 (73.28 °S, 114.97 °W; trap depth=410 m; water depth=830 m) was located in the central region of the Amundsen Sea Polynya. Samples were collected in this trap from 15 February 2012 to 20 February 2013. Station K3 (74.19 °S, 112.54 °W; water depth 1057 m) was located ~2 km north of the Dotson Ice Shelf inside the Amundsen Sea Polynya. The trap was deployed here at a depth of 490 m (567 m above the sea floor) and collected samples from 17 February 2012 to 1 March 2013. The trap moorings were equipped with RCM11 current meters placed 2 m below each trap. The flux and composition characteristics of small sinking particles collected at Stations K1, K2, and K3 are reported elsewhere in Kim et al. (2019).

Station K4 (73.89 °S, 118.72 °W; water depth 688 m) was located ~1.3 km off the East Getz Ice Shelf near Duncan Peninsula. The sediment trap at this location was deployed at a depth of 427 m from 1 February 2016 to 28 February 2018. The sampling intervals at Station K4 varied from 13 to 61 days, depending on the expected particle flux (Table 1). Each sample bottle was filled with filtered seawater collected from the same depth as the trap at the sampling site, with sodium borate buffer and 10 % formalin solution added, the latter as a preservative. Particle samples recovered from the traps were stored at 4 °C until analyzed. The particle flux was determined gravimetrically following removal of any conspicuous swimmers. The particulate organic carbon (POC) content was calculated as the difference between the total carbon content and the inorganic carbon content. The details of sample analysis are described in Kim et al. (2015, 2019).

Worm specimens were removed by hand from samples on return to the laboratory, and were stored in 10 % formalin solution. The specimen numbers were determined following visual assessment of whether each specimen was an entire body or a body part. The length and thickness of specimens were measured using a ruler and a Vernier caliper, respectively. At Station K4, juvenile scallops and a sea urchin were collected in addition to worms. The length of each of these organisms was measured using a ruler.

Five worm specimens were freeze dried, weighed, and finely ground using a mortar and pestle. The total carbon content of the ground samples (~0.3 mg) was determined using an elemental analyzer (vario MICRO cube; Elementar, Germany), with an uncertainty of 3 % relative standard deviation (RSD). The inorganic carbon content was determined using the same method, but based on ~0.2 mg samples that were ashed at 500 °C for 12 hours (Clarke et al., 1997; Obermüller et al., 2013). The inorganic fraction was negligible (0.1±0.2 % for carbon, and 0.1±0.1 % for nitrogen).

Sea ice concentration data were obtained from the European Centre for Medium-Range Weather Forecasts (ECMWF) ERA-Interim reanalysis (https://www.ecmwf.int), and are based on global coverage data (0.75°×0.75°) received daily from the Operational Sea Surface Temperature and Sea Ice Analysis (OSTIA) system.

.

## 3 Results

### 3.1 Temporal variation in sea ice concentration and fluxes of sinking particles and POC

Station K4 is located within the seasonal Amundsen Sea Polynya, and the sea ice concentration decreases to zero in summer (Figure 2). Ice-free conditions lasted for ~2.5 months from January to March 2016, and for ~4.5 months from late November 2016 to March 2017. In the other seasons the sea ice concentration was generally >85 %. The sinking particle flux varied from 11 to 978 mg m$^{-2}$d$^{-1}$, with a sampling duration-weighted average of 70 mg m$^{-2}$d$^{-1}$. Relatively high fluxes were found during the austral summer. However, the flux in the 2016−2017 summer was considerably lower than in the preceding summer. The POC flux varied from 1.0 to 37 mgC m$^{-2}$d$^{-1}$, with a sampling duration-weighted average of 5.3 mgC m$^{-2}$d$^{-1}$ (Figure 2; Table 1). The average POC and inorganic carbon contents were 9.2±4.0 % and 0.3±0.2 %, respectively (not shown).

### 3.2 Collection of invertebrate specimens

Long and slender worms were found in the sediment traps at Stations K1 and K3 during the 2012−2013 deployments. No worms were found in the traps at Station K2. The worm specimens were either entire bodies or body parts. The specimens were all similar in appearance (Figure 3). All had a rubbery texture and were easily torn apart when gently stretched. The bodies were round and slender and had a body thickness <5 mm. Species identification based on genetic methods was not successful, probably because the formalin preservative damaged the DNA (Macrogen Inc.; http://www.macrogen.com). Identification of the preserved specimens based on morphological characteristics also proved to be difficult because of the storage in formalin preservative for an extended time (>1 year; personal communication with Dr. Chernyshev Alexei

Viktorovich). Although it was not possible to identify the species, the interpretation of the results might be applicable to studies that involve the collection of any large benthic invertebrates.

In total, 33 specimens (including 18 incomplete bodies) were collected at Station K3 in 6 sampling bottles from April to August, and in mid-December 2012 (Figure 2, Table 1), and 24 specimens (10 complete and 14 incomplete bodies) were collected in July. Two specimens were collected at Station K1 in two sampling bottles in June 2012 and in early March 2013 (Figure 2). The lengths of the specimens collected at Stations K1 and K3 varied from 2 to 69 cm (average: 14±15 cm) (Figure 3).

The average total carbon and nitrogen contents of 5 randomly selected worm samples were 44.0±1.4 % and 7.7±1.1 %, respectively. The inorganic carbon content was negligible (average: 0.1 %). The POC content, expressed as the difference between the total and inorganic carbon contents, was 43.9±1.5 %. Considering the high POC content and presumably high protein content of these specimens, their bodies and/or gut content are likely to be mainly organic matter with a relatively small amount of ingested sedimentary material. The POC flux derived from worm specimens was estimated from the length and thickness of each specimen, the dry weight, the POC content, and the linear relationship between the volume of the specimen and the measured amount of POC (POC content in mg=0.052×volume of worm in mm$^3$+0.5498; R$^2$=0.94, $n$=5, $p$-value=0.0029). The estimated POC fluxes by worm specimens were 1.2 and 4.1 gC m$^{-2}$yr$^{-1}$ at Stations K1 and K3, respectively. One bottle, deployed for the month of June at Station K1, contained 590 mgC. At Station K3, large quantities of worms were found on two occasions: 1360 mgC in July and 590 mgC in August. The worm specimen flux corresponded to ~80 % and 500 % of the annual POC flux of sinking particles at Stations K1 and K3, respectively.

At Station K4 during the 2016–2018 deployment, there were found 11 juvenile scallops and 1 juvenile sea urchin in addition to 12 worm specimens (including 7 incomplete bodies). Worm specimens were found in 5 samples collected in March–September 2016 and June–July 2017 (Figure 2, Table 1). These worms were similar in shape to those collected at Stations K1 and K3 (Figure 3). One scallop was collected in August–September 2017, and 10 were collected in October–November 2017. The scallops were all juveniles and varied in size from 1.2 to 2.8 cm. The baffle of the sediment trap (approximately 2.5 cm diameter) would have blocked any larger creatures from falling into the trap. One juvenile sea urchin was also collected in October–November 2017. No specimens were collected under sea ice-free conditions.

## 4 Discussion

### 4.1 Potential mechanisms for transport of the benthic invertebrates

It is puzzling that large benthic invertebrates lacking swimming capability were collected in sediment traps deployed 130−567 m above the sea floor. The worms had no parapodia or appendages for swimming. It is unlikely that these organisms crawled up the mooring lines to the traps, and human activities were not responsible because the region is not accessible during winter, when most of the specimens were collected. According to the pressure registered to the current meters, ADCPs (Acoustic Doppler Current Profilers), and Microcats (Seabird, SBE-37SMP) moored with the sediment traps did not show any sign for

considerable tilting of the mooring lines to facilitate better access for the benthos (Kim et al., 2016). Several mechanisms may explain the transport of the collected organisms. Strong currents, especially near Station K3, may have been responsible for swiping and transporting these organisms to the trap sites. Stations K3 and K4 were within ~10 km from the nearest coast. Small juvenile scallops may be particularly affected by strong currents. In addition, scallops may use the current as a means of dispersal and translocation (Picken, 1980). The large size of the worms precludes the possibility that they were passively lifted but they may actively use the current. The collection of worms in April-August and the period of relatively strong current in July-September partly overlap. However, Station K1 was 200−500 km away from the coast where these worms presumably inhabit and are too remote for transport by current alone.

A likely explanation is that the organisms were transported by, and released from, sea ice. Two mechanisms could entrain benthic organisms in the sea ice: anchor ice formation and adfreezing of land fast ice. Ice foot formation, fast ice, and subsequent scouring are causes of disturbance of the benthic environment in the Antarctic (Barnes, 1999; Gutt, 2001; Barnes et al., 2011), and benthic organisms can be incorporated into the ice during these processes. For example, adfreezing by bottom fast ice can incorporate benthic organisms in addition to coarse sediment grains (Pfirman et al., 1990; Nürnberg et al., 1994). Several observations indicate that bottom fast ice formation is not a likely mechanism in this case. Firstly, the collected specimens were intact and well preserved, with no evidence of mechanical abrasion (Reimnitz et al., 1992); breaking of bottom fast ice would have destroyed these soft-bodied benthic organisms. Secondly, no coarse particles (grain size larger than sand) were collected in the sediment traps at the same time; small granules and rock fragments were collected only in one sample at Station K4 in April and May 2016. Thirdly, the organisms were collected mostly in winter; bottom fast ice at the shoreline would remain attached to the land during winter, and would thus only release entrained particles near the shore (Reimnitz et al., 1992). In contrast, anchor ice formation could have gently entrained these delicate organisms. Anchor ice forms in supercooled water (Dayton et al., 1969). Any anchor ice that formed on the bodies of benthic organisms would be buoyant, and thus able to lift them to the overlying sea ice canopy, in which they could become incorporated (Dayton et al., 1969; Heine et al., 1991). Organisms would be released when the attached anchor ice melted by the heat provided by the underlying water. Another hypothesis is that the benthic animals actually spend their juvenile period in a habitat underneath the sea ice and fall down to the seafloor. This idea was suggested by a reviewer, Dr. Paul Dayton, and we agree that this can be a possibility. This hypothesis is based on his visual inspections in numerous dives at McMurdo Sound that no baby nemerteans were observed and tiny sea urchins and scallops were rare. The undersurface of the sea ice can harbor a thick layer of frazil ice platelets formed by supercooled water, and cavities and tunnels formed by brine flow. Diatoms and other algae growing in and/or underneath the sea ice would supply food for juvenile sea urchins and pectins. Also amphipods and small invertebrates would provide food for juvenile nemertean recruits. These organisms may passively sink to the seafloor upon melting of the platelets or actively abandon the sea ice habitat due to depletion of algal food in the winter. This kind of habitat with large populations of these animals has not been observed yet and needs to be verified.

## 4.2 Potential for anchor ice formation in the Amundsen Sea

When ice grows on objects at the seafloor in supercooled water, it is called anchor ice (Denny et al., 2011). Anchor ice can grow on epibenthic animals as nucleation sites. Studies of anchor ice formation have focused mainly on sediment entrainment into sea ice in the Arctic (Pfirman et al., 1990; Reimnitz et al., 1992; Nürnberg et al., 1994), where this mechanism is considered to be the most important process for entrainment of sediment into sea ice and subsequent sediment transport (Nürnberg et al., 1994). Arctic sea ice transports well-preserved benthic organisms including mollusks and sea urchins, in addition to sediments (Reimnitz et al., 1992). Studies of anchor ice in the Antarctic have focused mostly on biological effects (Barnes, 1999).

Anchor ice has been reported in McMurdo Sound in the Ross Sea (Dayton et al., 1969; Robinson et al., 2014), Potter Cove in the South Shetland islands (Barrera-Oro and Casaux, 1990), and Ellis Fjord and a few other locations (Kirkwood and Burton, 1988 and references therein). Supercooling has not been reported in the Amundsen Sea because no hydrographic data near the coast are available for seasons other than summer. Observations in summer on the Amundsen Shelf have shown that winter water (water that mimics the properties of water formed in winter) was at near freezing temperature (potential temperature <−1.75 °C; Yager et al., 2016). Surface supercooling over polynyas is one mechanism for anchor ice formation (Mager et al., 2013). Strong winds, subfreezing temperatures, and intense turbulence in an open shallow sea are necessary for anchor ice formation (Reimnitz et al., 1992). Investigation of whether the polynyas in the Amundsen Sea during fall and winter have these conditions is a potential topic for future studies. The majority of the Amundsen Sea Polynya and the Pine Island Polynya close in the fall. However, strong katabatic winds generate narrow coastal leads and open spaces (Nihashi et al., 2015; Stammerjohn et al., 2015), where freezing predominates (Assmann et al., 2005).

Another mechanism for anchor ice formation is the so-called 'ice pump', which occurs underneath ice shelves (Mager et al., 2013). Lenard et al. (2014) investigated the potential for anchor ice formation in coastal Antarctic waters, focusing particularly on the ice pump mechanism. They estimated that a wide area of McMurdo Sound is suitable for anchor ice formation. The Amundsen Sea includes several ice shelves. According to a modeling study by Assmann et al. (2005), temperatures below surface freezing were simulated close to the Abbot and Getz Ice Shelves.

## 5 Implications and Conclusions

The observations show that transport and release of benthic invertebrates occurs episodically with potentially high interannual variability. No invertebrates were collected in sediment traps in the preceding year at either Station K1 or the central polynya (Ducklow et al., 2015; Kim et al., 2015). This phenomenon is episodic, probably because particular conditions must occur for anchor ice to form, if anchor ice is indeed the mechanism responsible (Kempema et al., 1989; Reimnitz et al., 1992). An implication for coastal benthic ecosystems is that anchor ice formation around Antarctica may be more prevalent than previously thought; the occurrence of many ice shelves and polynyas implies this possibility (Leonard et al., 2014; Mager et al., 2015).

Uplift of benthic organisms to the sea surface by anchor ice has been considered to be important for the Arctic food web (Reimnitz et al., 1992). The role of anchor ice in the transport of benthic organisms off the coast has not been investigated in the Antarctic. Organic carbon supplied by worms accounted for up to 5-fold the POC flux derived from primary production in the overlying water column. This supply of organisms occurred during winter, when little primary production occurs; hence, this flux may provide energy for overwintering organisms. Transport by anchor ice and sea ice rafting may be a means of dispersal for benthic invertebrates (Picken, 1980), provided that the organisms are able to survive this process. In this context, there is no evidence as to whether the organism entered the sediment traps alive or dead.

More studies are needed to clarify the spatial distribution of anchor ice formation. Lenard et al. (2014) noted that detailed information on ice shelf drafts, ground lines, and edges are needed to enable prediction of anchor ice formation, and observations of anchor ice are needed for validation of model outputs. Studies of these aspects of anchor ice formation in the Amundsen Sea would be a useful future study. In parallel, the impact of anchor ice on Antarctic biology, including as a disturbance to benthic ecosystems, a dispersal mechanism for benthic invertebrates, and as a supply of energy, nutrients, and detrital material to the deeper benthic ecosystems, are also needed to be investigated.

### Acknowledgement

Junsung Noh is thanked for carbon and nitrogen analyses of the worm specimens; Karen M. Assmann, Okhwan Yu, Dong-Hoon Im, Seung Hee Kim, and Eunah Han for helpful discussion; and the captain and crew of the IBRV *Araon* for their assistance during fieldwork. Special thanks are given to Drs. Chernyshev Alexei Viktorovich, In-Young Ahn, and Francyne Elias-Piera for their contributions to the identification of the worm specimens. We also thank Drs. Paul Dayton, Wei-Lei Wang, and Tiantian Tang for their constructive comments. This study made use of Rapid Response imagery from the Land, Atmosphere Near real-time Capability for EOS (LANCE) system, operated by the NASA/GSFC/Earth Science Data and Information System (ESDIS), with funding provided by NASA/HQ. This research was supported by the Korea Institute of Ocean Science and Technology (PN67330) and the Korea Polar Research Institute (PE18060). Minkyoung Kim was partly supported by the National Research Foundation of Korea grant, funded by the Korean Government (Global PhD fellowship 2015032018).

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

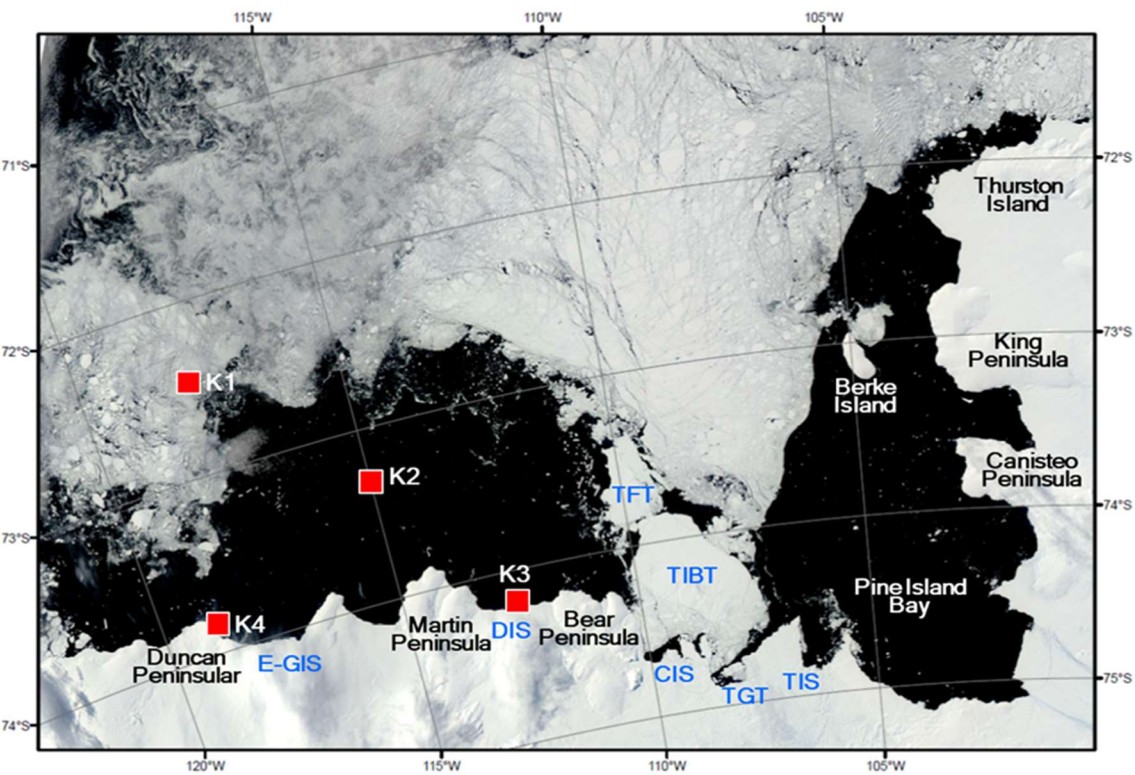

**Figure 1: Locations of sediment traps deployed in the Amundsen Sea. The satellite image showing the sea ice distribution was taken on 15 February 2012. The satellite image was obtained from Rapid Response imagery from the Land, Atmosphere Near real-time Capability for EOS (LANCE) system, operated by the NASA/GSFC/Earth Science Data and Information System (ESDIS). Areas marked from east to west: Thwaites Ice Shelf (TIS), Thwaites Glacier Tongue (TGT), Thwaites Iceberg Tongue (TIBT), Thwaites Fast-ice Tongue (TFT), Crosson Ice Shelf (CIS), Dotson Ice Shelf (DIS), and East Getz Ice Shelf (E-GIS).**

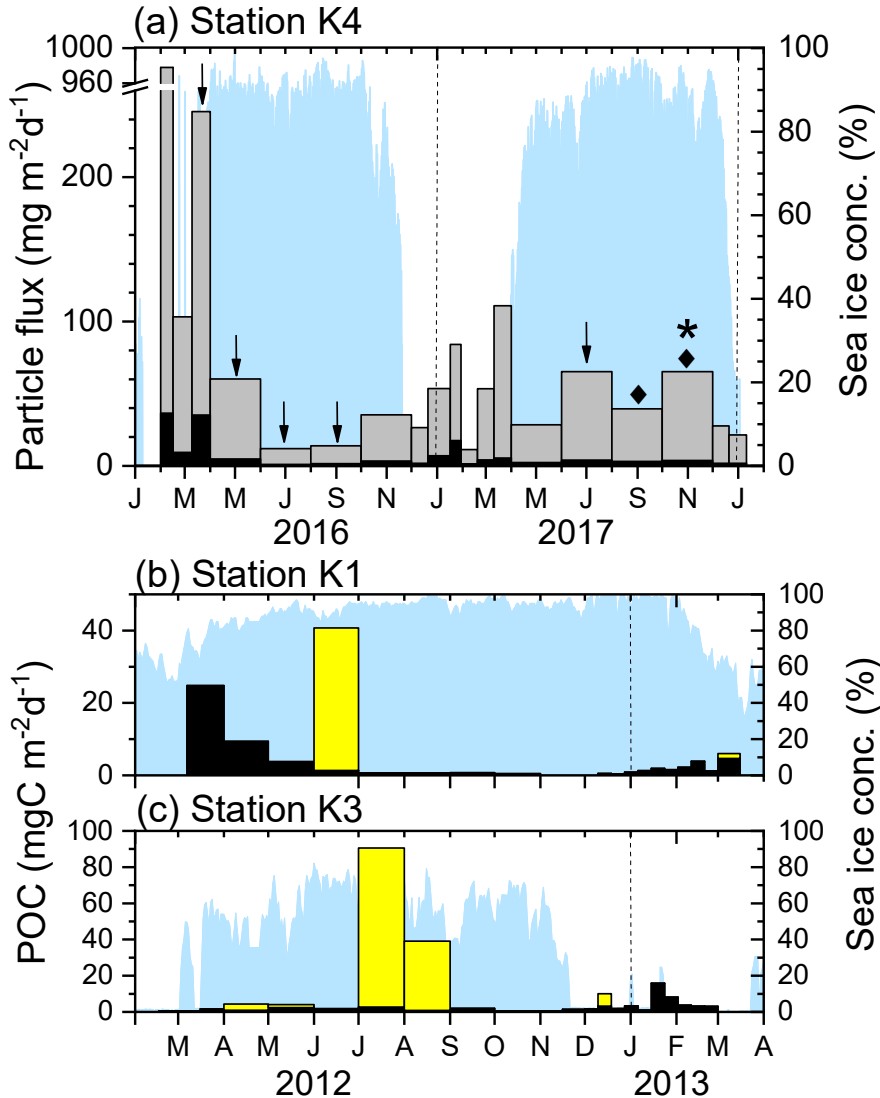

**Figure 2: (a) Fluxes of sinking particles (gray bars) and POC (black bars) at Station K4. The tick marks on the x-axis indicate the start of each month. The arrows, diamonds, and the star respectively indicate the periods when worms, scallops, and the sea urchin were collected. (b) POC fluxes by worms (yellow bars) and sinking particles (black bars) at Stations K1 and (c) K3. Note that the y-axis scales differ. The dotted vertical lines denote the beginning of each year. The blue shading indicates the sea ice concentration. POC flux and sea ice data for Stations K1 and K3 were redrawn from Kim et al. (2019).**

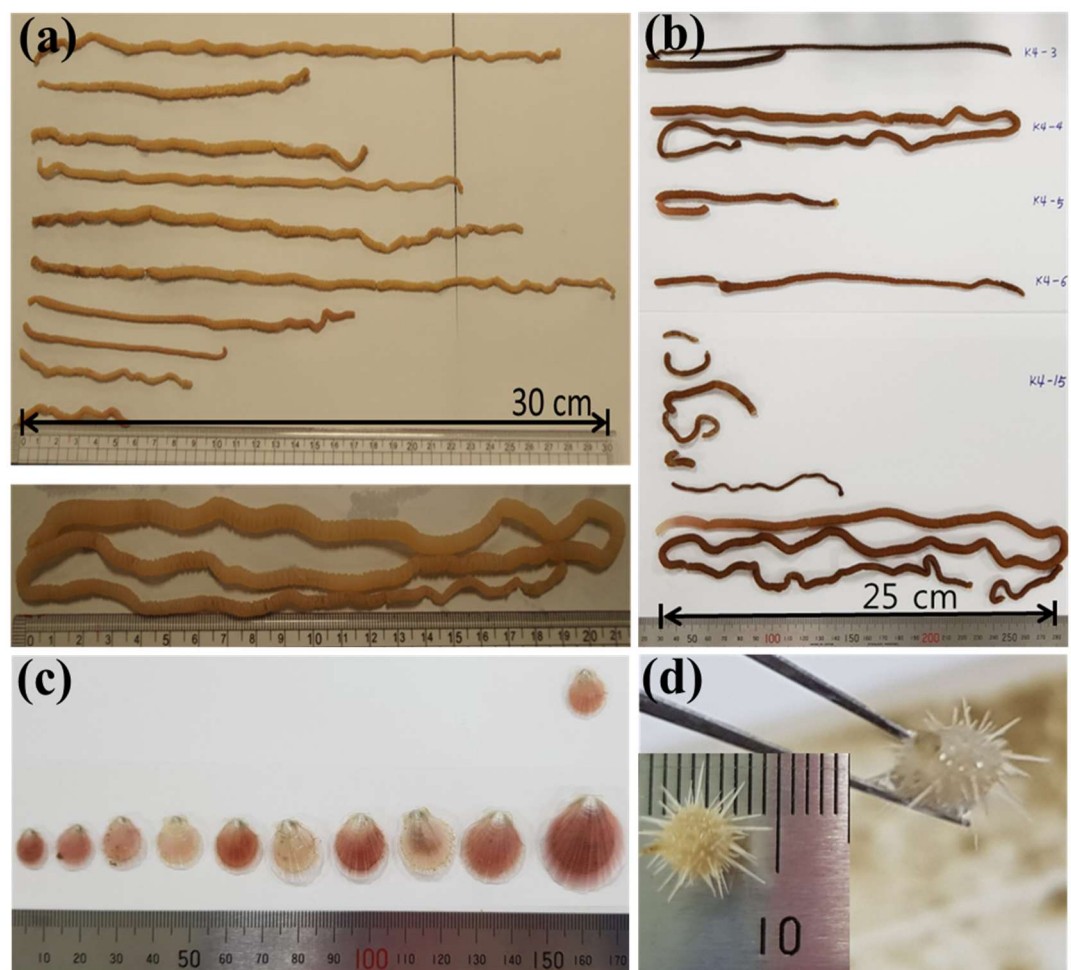

**Figure 3: (a) Worm specimens collected at Stations K1 and K3, and (b) at Station K4. (c) Scallops and (d) a sea urchin were collected at Station K4.**

**Table 1: Sequential trap schedules, and organisms collected at Stations K1, K3, and K4 in the Amundsen Sea. Quantity, lengths, and thicknesses of worm specimens collected in the sediment traps are shown. Total particle and POC fluxes are also presented for Station K4.**

| Cup # | Station K1 | | | | | Station K3 | | | | | Station K4 | | | | | | | |
|---|---|---|---|---|---|---|---|---|---|---|---|---|---|---|---|---|---|---|
| | Cup open date (mm/dd/yy) | Interval (days) | Number of whole body/part | Length (cm), thickness (mm) | Worm flux (mgC/bottle) | Cup open date (mm/dd/yy) | Interval (days) | Number of whole body/part | Length (cm), thickness (mm) | Worm flux (mgC/bottle) | Cup open date (mm/dd/yy) | Interval (days) | Total mass flux (mg m⁻² d⁻¹) | POC flux (mgC m⁻² d⁻¹) | Number of whole body/part | Length (cm), thickness (mm) | Worm flux (mgC/bottle) | Number of scallops/urchin |
| 1 | 3/7/12 | 25 | | | | 2/17/12 | 28 | | | | 2/1/16 | 37 | 978 | 37 | | | | |
| 2 | 4/1/12 | 30 | | | | 3/16/12 | 16 | | | | 2/16/16 | 23 | 103 | 94 | | | | |
| 3 | 5/1/12 | 31 | | | | 4/1/12 | 30 | 10 | (16,0.28) | 52 | 3/10/16 | 22 | 94 | 35 | | | | |
| 4 | 6/1/12 | 30 | 10 | (69,46) | 590 | 5/1/12 | 31 | 10 | (6.7,3.0) | 25 | 4/1/16 | 60 | 245 | 48 | 10 | (32.0,18) | 43 | |
| 5 | 7/1/12 | 31 | | | | 6/1/12 | 30 | 10 | (3.7,22) | 8 | 6/1/16 | 61 | 60 | 10 | 10 | (55.0,27) | 158 | |
| 6 | 8/1/12 | 31 | | | | 7/1/12 | 31 | 10/14 | note 1 | 1361 | 8/1/16 | 61 | 12 | 12 | 02 | note 3 | 33 | |
| 7 | 9/1/12 | 30 | | | | 8/1/12 | 31 | 1/4 | note 2 | 593 | 10/1/16 | 61 | 10 | 16 | 10 | (24.0,26) | 67 | |
| 8 | 10/1/12 | 31 | | | | 9/1/12 | 30 | | | | 12/1/16 | 61 | 14 | 34 | | | | |
| 9 | 11/1/12 | 31 | | | | 10/1/12 | 31 | | | | 1/7/17 | 35 | 16 | 35 | | | | |
| 10 | 11/16/12 | 15 | | | | 11/1/12 | 31 | | | | 1/30/17 | 13 | 54 | 70 | | | | |
| 11 | 12/1/12 | 15 | | | | 11/16/12 | 15 | | | | 1/17/17 | 84 | 18 | 18 | | | | |
| 12 | 12/10/12 | 9 | | | | 12/1/12 | 15 | | | | 1/30/17 | 11 | 11 | 14 | | | | |
| 13 | 12/19/12 | 9 | | | | 12/10/12 | 9 | 10 | (14.0,23) | 31 | 2/19/17 | 20 | 54 | 42 | | | | |
| 14 | 12/28/12 | 9 | | | | 12/19/12 | 9 | | | | 3/11/17 | 21 | 111 | 54 | | | | |
| 15 | 1/6/13 | 9 | | | | 12/28/12 | 9 | | | | 4/1/17 | 61 | 29 | 24 | 2/5 | note 4 | | 281 |
| 16 | 1/15/13 | 9 | | | | 1/6/13 | 9 | | | | 6/1/17 | 61 | 65 | 40 | | note 5 | | 10 |
| 17 | 1/24/13 | 9 | | | | 1/15/13 | 9 | | | | 8/1/17 | 61 | 39 | 32 | | note 5 | | 101 |
| 18 | 2/2/13 | 9 | | | | 1/24/13 | 9 | | | | 10/1/17 | 65 | 3.7 | 19 | | | | |
| 19 | 2/11/13 | 9 | | | | 2/2/13 | 9 | | | | 12/1/17 | 20 | 28 | 19 | | | | |
| 20 | 2/20/13 | 9 | (25,23) | | 6 | 2/11/13 | 9 | | | | 1/22/17 | 21 | 21 | 18 | | | | |
| 21 | 3/1/13 | 15 | | | | 2/20/13 | 9 | | | | 1/11/18 | 21 | | | | | | |
| | | | | | | 2/1/18 | 27 | | | | 1/18/18 | 27 | | | | | | |

note 1: The length in cm and thickness in mm of the 24 specimens collected in cup #6 were (31.5, 4.1), (16.5, 3.5), (19.5, 4.4), (24.5, 3.0), (27.7, 3.8), (31.5, 2.4), (17.5, 2.4), (11.0, 2.2), (9.0, 2.3), (6.0, 2.5), (2.0, 5.1), (3.5, 4.9), (7.9, 3.8), (4.0, 3.5), (1.5, 4.4), (4.8, 3.8), (8.2, 4.6), (2.0, 3.0), (10.0, 4.4), (9.0, 3.9), (5.3, 4.3), (3.5, 3.3), (20.4, 2.9), (3.0, 2.0).

note 2: The length in cm and thickness in mm of the 5 specimens in cup #7 were (22.7, 3.4), (53.5, 4.1), (11.8, 4.0), (11.4, 2.2), (7.7, 2.1).

note 3: The length in cm and thickness in mm of the 5 specimens in cup #5 were (4.0, 2.9), (12.0, 1.9).

note 4: The length in cm and thickness in mm of the 7 specimens in cup #15 were (2.0, 1.6), (5.0, 1.5), (8.0, 2.9), (7.0, 1.5), (3.0, 2.6), (12.0, 0.2), (80, 2.7).

note 5: The height and width in cm of the 11 scallops collected in cup #16 and #17 were (1.3, 1.3), (1.2, 0.9), (1.3, 1.1), (1.5, 1.4), (1.5, 1.4), (1.6, 1.5), (1.7, 1.6), (1.9, 1.7), (1.9, 1.6), (2.0, 2.0), (2.8, 2.5). The diameter of the sea urchin was approximately 0.5 cm excluding the spines.