# Peer review of "Collection of large benthic invertebrates in sediment traps in the Amundsen Sea, Antarctica"

_Biogeosciences, 2019_

## Referee Comment (RC1) · Anonymous Referee #1 · 15 Mar 2019

This is an amazing paper. I have spent the weekend thinking about it and I think I have come up with an alternative hypothesis.

But first, the paper has to be published with or without my ideas. Indeed, while I don't agree with their anchor ice hypothesis, it is the only reasonable explanation in the literature. Moreover, it is the second best review of anchor ice that I have seen, and I described anchor ice and stalactites in the 19060s, so know something about the issue. Their observations and explanation really deserve to be published. The paper could be accepted absolutely as it is because the observations are extremely interesting and valuable to all polar marine scientists and their detailed well presented explanation is

the only one consistent with the published literature that they review VERY completely. So in my mind you can accept it as it is, but then let me have an opportunity to present my alternative highly speculative explanation in the same issue because I think that it might be correct!

Or, alternatively, allow them to make the decision and perhaps work my thinking into their paper. Whatever you wish is fine with me, but it is important to get the paper out to the community. I will reply to your questions below and attach my review. Thank you for your patience getting me this interesting manuscript.

Please also note the supplement to this comment:
https://www.biogeosciences-discuss.net/bg-2019-57/bg-2019-57-RC1-supplement.pdf

**Supplement:**

Review of MS by Kim et al: "Collection of large benthic invertebrates …."

I think that this is an amazing observation and absolutely deserves to be published quickly.

Outside of the Mager papers I was not aware of until asked to review this, I think that these authors have put together the most scholarly and complete discussion of anchor ice I have ever seen. I am still struggling to understand how those animals got there and will talk about some ideas, but the important thing is to get the paper published so the general community can discuss it. I cannot emphasize strongly enough that I consider it important irrespective as to whatever the ultimate explanation turns out to be, this represents a very unusual situation that will much enhance our understanding of polar marine ecosystems.

The following comments are not meant to be criticisms that stand in the way of publication, but just my efforts to understand the processes that lead to these animals falling into the traps. I do think that the issues I discuss merit some discussion in the text if only to focus other scientists on the issue of dispersal to the trap area and the density of invertebrates transported to the area above the traps. Also, I offer an alternative hypothesis that has never been considered before, but I think it merits some sort of mention in the paper.

First, these are shallow water species assuming they are *Parborlasia corrugatus* (the worm), *Sterechinus neumayeri* (the urchin), and *Adamussium colbecki* (the scallop). I think I can assert that they were not uplifted from deep water near the traps, but rather were sourced in shallow water far from the traps.

The traps are small, 80 cm aperture diameter, but still extremely tiny compared to the ocean bottom! Anchor ice in my McMurdo experience does lift animals, but not many and the slimy ones such as nemerteans or anemones fall out of the ice quickly, and I would not expect to see nemerteans transported very far, if at all. So, my own personal experience with anchor ice would agree that it is possible for a very few tiny pectin or urchins to be transported out to sea, but relatively few. It would be an amazing put possible coincidence for anchor ice to carry an urchin or pectin and have it fall in the trap. But a single one arriving this way would be astounding. You have many. The probability of anchor ice getting so many shore-based animals so far to sea falling into a tiny orifice is remote beyond calculation.

I offer an alternative explanation for which there is absolutely no evidence (but nor is there any evidence before this paper of such transportation). These are very small immature animals that have just settled and metamorphosed. In almost 1,000 dives at McMurdo I have never seen any baby nemerteans, none. If those are Parborlasia, this in itself is extraordinary to see so many baby nemerteans. And that many tiny urchins and scallops are also rare in the natural situations I have seen. I offer the following hypothesis that might explain so many very young animals falling into the traps.

A lot of light comes through thin annual sea ice and there are dense growths of diatoms and other algae on the undersurface of the ice. Sometimes these diatoms and *Phaeocystis* chains

hang down from the ice.  Now assume that there is a thick layer of frazil ice platelets formed by upwelling of super cooled water.  Furthermore, brine flows out from annual sea ice making stalactites and leaving large cavities and tunnels in the sea ice that become rich algal farms that are colonized by amphipods and small fish (at McMurdo *Pagothenia borchgrevinki*).  Assuming that this happened in your area, and the entire "roof" above your traps had all this biological activity, it is unheard of but possible for the pelagic larvae of the invertebrates that you recovered to have experienced a very heavy recruitment in this "farm" of algal food (for the urchins and pectins; the amphipods and other tiny metamorphosized invertebrates would be food for the nemertean recruits to have eaten.  Now you have large populations of the same animals you trapped above your traps.  So assume that the surface waters warmed and the platelets melted , and your Amundsen Sea might have large numbers of invertebrates sinking to the bottom and falling into your traps.

I actually think that this is a better explanation than anchor ice uplift and transport and I suggest that you find a way to work it into your paper.  But again, absolutely, the paper deserves to be accepted one way or another as it is a very unusual and interesting observation.

Good luck,

Paul Dayton

---

## Author Comment (AC1) · 27 Mar 2019

We appreciate the positive and really constructive review. We suggested lift of benthos by anchor ice formation, incorporation into the overlying sea ice canopy, and subsequent ice rafting as the most plausible process for the unexpected collection of the benthic organisms in the sediment traps. We have not considered the mechanism Dr. Dayton suggested especially for the case of scallops and sea urchins. We agree that growth of worms underneath the sea ice and falling to the seafloor may be another possible mechanism. We believe this idea can be successfully incorporated into our paper in the revision.

---

## Referee Comment (RC2) · Weilei Wang (Referee) · 6 May 2019

This paper reports that benthic invertebrates with no swimming ability were collected in sediment traps positioned well above the sea floor. The authors suggest that anchor sea ice formation, transport, and following release is the possible mechanism to explain the occurrence, which is reasonable to me. They also suggest that this kind of transport could be an important carbon source for the Antarctic food web, and a means of dispersal for benthic invertebrates.

Overall, this paper is well-written, and the data are valuable to the community. I suggest for publication after some very minor changes.

[Figure]

**Specific comments**

1. A short introduction about anchor sea ice will be beneficial for broad readers.

2. I do not agree to use the low average current speeds as an evidence for weak current. Between June and September, the current speeds at K3 were much higher than the average current speed (Fig. 3c of Kim et al., 2019 J. Mar. Syst.). This is also the time when most worms were collected (Fig. 2c). You should discuss possible connections between current speed and worm collection.

3. What is the bathymetry of the studied area? How far are the sampling sites from the nearest continent? Is it possible that the worms were from shallow water sediment and transported by current to the traps. As also been suggested in Kim et al., (2019), an average current speed of 10 cm/s and sinking speeds between 10-100 m/d together could enable traps to collect particles originated tens of kilometers away from the trap site.

4. Were the traps tilted? Did you have a tilt sensor on the trap? Even if the current speeds 2 m below the traps were not high, the whole mooring system could also be tilted. Do sediment trap positions help explain the reported collection?

5. p.3 l.8 '...from October 2016 to March 2017' is not consistent to what is shown in Fig. 2a.

Best of luck!
Wei-Lei Wang

---

## Referee Comment (RC3) · Tiantian Tang (Referee) · 10 May 2019

The manuscript by M. Kim describes a discovery of benthic invertebrates from the sediment traps deployed in Amundsen Sea, Antarctica. They discussed several possibilities and concluded that the dispersal by anchor ice seems to be a plausible reason for this discovery. The redistribution of geochemically important components like iron by sea ice has been proposed as an extra source of trace nutrients for those HNLC region of Southern Ocean. While this paper suggests another mechanism to redistribute benthic materials to the ocean surrounding Antarctica. This mechanism potentially is a more efficient way to influence the geochemical cycling in Antarctica region than sea ice, in

which the nutrients mostly comes from atmospheric deposition. This paper is generally well organized and discussed in a clear logic. The topic fully meets the scopes of Biogeosciences. However, a revision of this manuscript is suggested with the following comments addressed before it can be published.

1) My major concern is the introduction part. Some detailed review should be provided on sea ice or anchor ice transport and their importance to Antarctica environments. The environmental setting of Amundsen sea should also be introduced with information like seasonal variations in nutrient supply, primary productivity, export productivity and so on. Benthic community structure in Antarctic coastal water should also be discussed if available, particularly those related to those invertebrates.

2) Another suggestion is a broader implication of benthic materials dispersal by anchor ice for Antarctic ecosystem. I noticed the author discussed the contribution of those benthic organisms vs POC flux from primary production. Then how about the cycling of other benthic components like nutrients, detrital materials and so on? Their geochemical importance under a changing climate might be as importance as the sea ice melting. I agree with the author that more future work should be addressed on how this new mechanism influences the Antarctic ecosystem, which of course could not be fully covered in this manuscript just considering the difficulty to work in Antarctica.

3) Page3, line18, genetic identification is very challenge even without formalin, since the organism has been frozen for a very long time before captured in trap. I suggest to identify those species with some traditional taxonomic approach.

4) Page3, line 29. The worms are sediment scavenger. They ingest the sediment as a whole, and their guts contain large abundance of digested sediment, which is reshaped both physically and chemically. Therefore, the worm not only contributes to the POC as biomass, but also the sediment materials. Anyway, my point here is the contribution of zooplankton and other higher level organisms has long been ignored in the evaluation of POC flux with current strategy of trap collection. If we count in all the materials in

the cup, the contribution of worms might be not that astonishingly high.

---

## Author Comment (AC2) · 15 May 2019

We appreciate the kind and constructive review. We will incorporate all specific comments raised by reviewer in the revision.

1. Short introduction about anchor ice

R) We will add more details about the anchor ice in the revision.

2. Transport by current?

R) We will add some more discussion about this point in the revision. It is possible that some organisms are transported by current passively and/or they may use current

actively for transportation/dispersal. Worms were collected from April to September and the strong current was mainly observed from July to September. Even though timings for benthos collection and strong current do not match perfectly, we agree with the reviewer that current can be potentially important.

3. Bathymetry, distance?

R) We will add information about bathymetry and distance from coastline in more detail. Sediment traps at Stations K3 and K4 were ∼2 km and ∼1.3 km away from the nearest peninsula. In the case of Station K1, it is 200∼500 km away from the coastline. So in this case, transport by current from the coastline does not seem feasible.

4. Sediment trap tilting

R) The traps were not equipped with tilt sensors. However, the pressure registered by a MicroCAT (Sea-Bird Electronics, SBE37SM-RS232) provides indirect information on the vertical position of the traps. For example, at station K3 the pressure fluctuated daily with an amplitude of <1.6 dbar (MicroCAT moored at 490 m) because of the tide and did not show any out-of-phase signal caused by tilting of the mooring line. Considerable titling of the whole mooring line would be necessary to position the traps near the seafloor for allowing benthos to reach. That kind of change in sediment trap position was not observed in the pressure monitoring.

5. Sea ice concentration.

R) It will be corrected correspondingly.

---

## Author Comment (AC3) · 15 May 2019

We appreciate the review. We will incorporate all specific comments raised by reviewer in the revision.

1.Detailed reviews on sea ice or anchor ice transport, their importance to the Antarctic

R) We will add more relevant lines of information in the revision as suggested.

2.Cycling of other components?

R) Cycling of nutrients, detrital materials through the ice rafted transport are important. The role of anchor ice in transporting sediment particles has not been studied

or reported in the Antarctic, in contrast to the Arctic where this phenomenon has been reported to be a main mechanism for sediment particle entrainment into sea ice and dispersal. We will include the reviewer's suggestion in the revision as a potentially important future research.

3.Traditional taxonomic approach

R) In addition to the genetic tool, we have also put effort for species identification based on conventional approaches asking benthos experts. Unfortunately, we have not been successful. For example, Dr. Chernyshev Alexei Viktorovich in Russia provided his opinion. The consensus was that specimens preserved in formalin for an extended time period (> 1 year) are very difficult to identify.

4.Gut content and contribution of zooplankton

R) Reviewer's point regarding the gut content is of importance. We have not tried to separate the gut content for further analysis. However, considering the high organic carbon content (44 %) and presumably high protein content of these specimens, their bodies are likely to be mainly organic matter with relatively small amount of sediment material. The sinking particles contain high content of non-biogenic material (or lithogenic material) supplied from sediment resuspension to begin with. Conventionally zooplanktons (so called "swimmers") collected in sediment traps are not considered as a part of passive particle flux. If the collected benthos in our study were able to swim actively to reach the traps, they should not be included in sinking particle flux. We only wanted to make a point that the carbon flux by them can be potentially important considering the small sinking POC flux.

---

## Author Response (AR1)

Dear Editor,

Thank you for considering our paper for publication in *Biogeosciences*. We have revised the manuscript incorporating virtually all comments provided by the reviewers during the open discussion.

Mainly, an alternative hypothesis suggested by Reviewer #1 has been added. Also the potential role of current is included as suggested by Reviewer #2. All minor comments have been incorporated.

There was a minor mistake in the results of the manuscript. We did not include the area of the trap opening ($0.5 \text{ m}^2$) in the calculation of total particle flux and POC flux at Station K4. In the revision, this mistake has been corrected. Specifically, we corrected Figure 1a (simply, the left y-axis scale was doubled) and the flux values in Table 1, and a few places in the result section where the flux values were mentioned. We believe that this correction does not affect the main content and claims made in the paper.

We hope that our revision has successfully incorporated all comments raised by reviewers.

If there are any additional comments or questions, please contact Minkyoung Kim, who can be reached by email, mini324@snu.ac.kr.

We appreciate again your effort and time.

Best regards,

Minkyoung Kim and co-authors

**Response to reviewers' comments.**

**Reviewer 1**

The alternative hypothesis suggested by the reviewer is added in the discussion. It reads as follows:

"Another hypothesis is that the benthic animals actually spend their juvenile period in a habitat underneath the sea ice and fall down to the seafloor. This idea was suggested by a reviewer, Dr. Paul Dayton, and we agree that this can be a possibility. This hypothesis is based on his visual inspections in numerous dives at McMurdo Sound that no baby nemerteans were observed and tiny sea urchins and scallops were rare. The undersurface of the sea ice can harbor a thick layer of frazil ice platelets formed by supercooled water, and cavities and tunnels formed by brine flow. Diatoms and other algae growing in and/or underneath the sea ice would supply food for juvenile sea urchins and pectins. Also amphipods and small invertebrates would provide food for juvenile nemertean recruits. These organisms may passively sink to the seafloor upon melting of the platelets or actively abandon the sea ice habitat due to depletion of algal food in the winter. This kind of habitat with large populations of these animals has not been observed yet and needs to be verified."

**Reviewer 2**

Short introduction about anchor ice

- We have added a few sentences in the introduction and also in the discussion.

Transport by current?

- Agreeing with the reviewer, we have added the following in the discussion: "Strong currents, especially near Station K3, may have been responsible for swiping and transporting these organisms to the trap sites. Stations K3 and K4 were within ~10 km from the nearest coast. Small juvenile scallops may be particularly affected by strong currents. In addition, scallops may use the current as a means of dispersal and translocation (Picken, 1980). The large size of the worms precludes the possibility that they were passively lifted but they may actively use the current. The collection of worms in April-August and the period of relatively strong current in July-September partly overlap. However, Station K1 was $200-500$ km away from the coast where these worms presumably inhabit and are too remote for transport by current alone."

Information on bathymetry and distance from the coast

- Distance of the mooring sites from the nearest coast has been added as follows:

"Stations K3 and K4 were within ~10 km from the nearest coast."

"Station K1 was 200−500 km away from the coast where these worms presumably inhabit and too remote for transport by current alone."

Sediment trap tilting

- The traps were not equipped with tilt sensors. However, the pressure registered by Microcats (Seabird Electronics, SBE37SM-RS232), current meters, and ADCPs provides indirect information on the vertical position of the traps. For example, at station K3 the pressure fluctuated daily with an amplitude of <1.6 dbar (RCM moored at 490 m) because of the tide and did not show any out-of-phase signal caused by tilting of the mooring line. Considerable titling of the whole mooring line would be necessary to position the traps near the seafloor for allowing benthos to reach. That kind of change in sediment trap position was not observed in the pressure monitoring. We have added the following:

"According to the pressure registered to the current meters, ADCPs (Acoustic Doppler Current Profilers), and Microcats (Seabird, SBE-37SMP) moored with the sediment traps did not show any sign for considerable tilting of the mooring lines to facilitate better access for the benthos (Kim et al., 2016)."

Timing for sea ice concentration reduction.

- The timing for sea ice reduction has been corrected from October to late November.

**Reviewer 3**

Adding detailed reviews on sea ice or anchor ice transport, their importance to the Antarctic, environmental setting of the Amundsen Sea etc. in the introduction.

- We have added a paragraph for general introduction of the Amundsen Sea. There are already several good review papers stemming from major field campaigns. Therefore, instead of providing detailed review of the Amundsen Sea biogeochemistry, we opted to add a short general introduction with relevant references for the interested reader. In addition, intense review on anchor ice is already provided in the discussion section. The following paragraph has been added in the introduction.

"The majority of the Amundsen Shelf in the Antarctic is perennially covered with sea ice, except for the two seasonal polynyas. The Amundsen Sea polynya in the west Amundsen Sea is the most productive polynya around Antarctica (Arrigo and van Dijken, 2003). Intensive flux of particulate organic carbon to the seafloor occurs in the austral summer while the sea interior is in starvation in the other seasons (Ducklow et al., 2015; Kim et al., 2015; Kim et al., 2019). Biogeochemical processes related to biological pump in the Amundsen Sea have been investigated by recent field campaigns (Arrigo and Alderkamp, 2012; Yager et al., 2012; Meredith et al., 2016; Lee et al., 2017)."

"The distinct environmental conditions and characteristics of the polar seas relative to temperate and tropical oceans probably explain the unusual occurrence of benthic invertebrates in sediment traps. For example, starvation in the winter due to a reduced supply of organic matter from the overlying water column may stimulate the relocation of benthos. The undersurface of the sea ice may provide a habitat for juvenile benthos before they settle to the seafloor. Anchor ice, which forms at the seafloor in supercooled water, can lift benthos to the overlying sea ice for further transport by ice rafting (Dayton et al., 1969)."

Broader implication of benthic material dispersal by anchor ice for Antarctic ecosystem
- Dispersal of organic matter, nutrients, and detrital materials through anchor ice formation and ice rafting can potentially be very important as the reviewer pointed out. Unfortunately, the role of anchor ice in transporting sediment particles not to mention benthic animals has not been studied or reported in the Antarctic, in contrast to the Arctic where this phenomenon has been reported to be a main mechanism for sediment particle entrainment into sea ice and dispersal. Our paper is the first one suggesting the potential importance of anchor ice formation in the Antarctic in this sense. We hope that more studies focus on this topic in the future and our paper provides a seed for that.

Traditional taxonomic approach
- In addition to DNA analysis, we have also put effort for species identification based on conventional approaches asking benthos experts. Unfortunately, we have not been successful. For example, Dr. Chernyshev Alexei Viktorovich in Russia provided his opinion. But the general consensus was that specimens preserved in formalin for an extended time period (> 1 year) are very difficult to identify.

Gut content and contribution of zooplankton

- We have not examined the gut content unfortunately. We only measured POC content of the specimens, which was about 44 %. Based on this we believe that the specimen is mainly organic matter. It is definitely possible that they contain some sedimentary particles.

- Regarding the zooplankton collection in sediment traps, we observed a few krills. The reviewer is right that these large zooplanktons can form a significant portion of particle flux. However, discussion whether to include zooplankton as a part of particle flux or not is beyond the focus of our paper.

[revised manuscript text omitted]

**Station K3**

| Cup open date interval (days) | Length (cm)/thickness body/gut (mm) | Number of whole body/gut thickness | Worm flux (mg C/hole) | Total mass flux (mg/m²/d) | POC flux (mg/m²/d) |
|---|---|---|---|---|---|
| 2/7/12 | 28 | | | 15 | 37 |
| 3/16/12 | 16 | | | 23 | 94 |
| 4/7/12 | 30 | 16 | | 22 | |
| 5/7/12 | 30 | 10 (16.0,28) | 52 | 60 | 35 |
| 6/7/12 | 31 | 10 (67,3.0) | 25 | 61 | 48 |
| 7/7/12 | 31 | 10 (3.7,2.2) | 8 | 61 | 10 |
| 8/7/12 | 31 | | | 61 | 14 |
| 9/7/12 | 30 | 14 note1 | 136l | 61 | 35 |
| 10/7/12 | 31 | 10/4 note2 | 593 | 61 | 16 |
| 11/7/12 | 31 | | 31 | | |
| 11/16/12 | 15 | | | | |
| 12/1/12 | 9 | | | | |
| 12/10/12 | 9 | 10 (14.0,2.3) | | | |
| 12/19/12 | 9 | | | | |
| 12/28/12 | 9 | | | | |
| 1/6/13 | 9 | | | | |
| 1/15/13 | 9 | | | | |
| 1/24/13 | 9 | | | | |
| 2/2/13 | 9 | | | | |
| 2/11/13 | 9 | | | | |
| 2/20/13 | 9 | | | | |

**Station K4**

| Cup # | Cup open date interval (days) | Length (cm)/thickness body/gut (mm) | Number of whole body/gut thickness | Length (cm)/thickness scallops (mg C/hole) | Number of scallops/urchin |
|---|---|---|---|---|---|
| 5 | 7/1/12 | 31 | | | |
| 6 | 8/1/12 | 31 | 10 (3.7,2.2) note 1 note 2 | | |
| 7 | 9/1/12 | 30 | 14 | | |
| 8 | 10/1/12 | 31 | | | |
| 9 | 11/1/12 | 15 | | | |
| 10 | 11/16/12 | 15 | | | |
| 11 | 12/1/12 | 15 | | | |
| 12 | 12/10/12 | 9 | 10 (14.0,2.3) | | |
| 13 | 12/19/12 | 9 | | | |
| 14 | 12/28/12 | 9 | | 02 (32.0,18) 43 | 281 |
| 15 | 1/6/13 | 9 | 10 (24.0,26) 67 | (55.0,27) 158 | |
| 16 | 1/15/13 | 9 | | note 3 33 | |
| 17 | 1/24/13 | 9 | | note 5 10 | 10l |
| 18 | 2/2/13 | 9 | | note 5 | |
| 19 | 2/11/13 | 9 | | note 4 281 | |
| 20 | 2/20/13 | 10 | 10 (24.0,26) | | 10l |
| 21 | 3/1/13 | 15 | (2.5,2.3) 6 | 67 33 | |

note 1: The length in cm and thickness in mm of the 24 specimens collected in cup #6 were (31.5, 4.1), (16.5, 3.5), (19.5, 4.4), (24.5, 3.0), (27.7, 3.8), (31.5, 2.4), (17.5, 2.4), (11.0, 2.2), (9.0, 2.3), (6.0, 2.5), (2.0, 5.1), (3.5, 4.9), (7.9, 3.8), (4.0, 3.5), (1.5, 4.4), (4.8, 3.8), (8.2, 4.6), (2.0, 3.0), (10.0, 4.4), (9.0, 3.9), (5.3, 4.3), (3.5, 3.3), (20.4, 2.9), (3.0, 2.0).

note 2: The length in cm and thickness in mm of the 5 specimens in cup #7 were (22.7, 3.4), (53.5, 4.1), (11.8, 4.0), (11.4, 2.2), (7.7, 2.1).

note 3: The length in cm and thickness in mm of the 5 specimens in cup #5 were (4.0, 2.9), (12.0, 1.9).

note 4: The length in cm and thickness in mm of the 7 specimens in cup #15 were (2.0, 1.6), (5.0, 1.5), (8.0, 2.9), (7.0, 1.5), (3.0, 2.6), (12.0, 0.2), (80, 2.7).

note 5: The height and width in cm of the 11 scallops collected in cup #16 and #17 were (1.3, 1.3), (1.2, 0.9), (1.3, 1.1), (1.5, 1.4), (1.5, 1.4), (1.6, 1.5), (1.7, 1.6), (1.9, 1.6), (2.0, 2.0), (2.8, 2.5). The diameter of the sea urchin was approximately 0.5 cm excluding the spines.